# Effects of In Vivo Intracellular ATP Modulation on Neocortical Extracellular Potassium Concentration

**DOI:** 10.3390/biomedicines10071568

**Published:** 2022-07-01

**Authors:** Azin EbrahimAmini, Bojana Stefanovic, Peter L. Carlen

**Affiliations:** 1Krembil Research Institute, Toronto, ON M5T 0S8, Canada; carlen@uhnresearch.ca; 2Institute of Biomedical Engineering, University of Toronto, Toronto, ON M5S 3G9, Canada; 3Sunnybrook Health Sciences Center, Medical Biophysics, Toronto, ON M4N 3M5, Canada; bojana@sri.utoronto.ca; 4Departments of Medicine and Physiology, University of Toronto, Toronto, ON M5S 1A8, Canada

**Keywords:** extracellular potassium concentration, intracellular ATP concentration, Na/K ATPase pump, ATP-sensitive potassium channel

## Abstract

Neuronal and glial activity are dependent on the efflux of potassium ions into the extracellular space. Efflux of K is partly energy-dependent as the activity of pumps and channels which are involved in K transportation is ATP-dependent. In this study, we investigated the effect of decreased intracellular ATP concentration ([ATP]i) on the extracellular potassium ion concentration ([K]o). Using in vivo electrophysiological techniques, we measured neocortical [K]o and the local field potential (LFP) while [ATP]i was reduced through various pharmacological interventions. We observed that reducing [ATP]i led to raised [K]o and DC-shifts resembling spreading depolarization-like events. We proposed that most likely, the increased [K]o is mainly due to the impairment of the Na/K ATPase pump and the ATP-sensitive potassium channel in the absence of sufficient ATP, because Na/K ATPase inhibition led to increased [K]o and ATP-sensitive potassium channel impairment resulted in decreased [K]o. Therefore, an important consequence of decreased [ATP]i is an increased [K]o. The results of this study acknowledge one of the mechanisms involved in [K]o dynamics.

## 1. Introduction

Information travels along and between neurons via action potentials. An action potential is a wave of an electrical discharge, an interplay between sodium ions (Na) and potassium ions (K) which is dependent on Na and K channels/pumps some of which are dependent on energy (ATP) to function properly [1,2]. The brain comprises ~2% of a human’s body weight but consumes ~20% of the body’s total energy budget [3]. Without a doubt, disruption of [ATP]i must have consequences for cellular functionality. In this study, in particular we are interested in the effect of [ATP]i on the [K]o. To investigate this topic, we have measured the [K]o and LFP while modulating [ATP]i using 2,4-dinitrophenol (DNP), Endothelin-1 (ET-1), 2-Deoxyglucose (2-DG), Oubain and Glibenclamide.

DNP is a weak mitochondrial uncoupling agent relative to other uncoupling agents (i.e., FCCP, CCCP). Mitochondrial uncoupling leads to a disrupted proton gradient and oxidative phosphorylation [4]. Protons that could have been used for ATP synthesis are instead leaked across the mitochondrial membrane leading to lower intracellular ATP production [4]. It is known that DNP initiates spreading depolarization (SD) and inhibits K active transport from the extracellular space to intracellular space [5]. Local application of DNP on cortex elicited SD similar to KCl application [6]. Application of 0.2 mM DNP on fibers recovering from 5–10 min stimulation significantly reduced K influx [7].

ET-1 as one of the most potent vasoconstrictors and neuronal/astroglial modulator alters endothelial cells and vascular smooth muscle [8]. ET-1-induced ischemia causes local oxygen and glucose deprivation, leading to lower [ATP]i production [8,9]. Dreier et al. (2002) demonstrated a small and slow increase in [K]o before seeing SD and its concomitant sharp [K]o rise [8]. This kind of small rise in [K]o has been also seen before peri-infarct depolarizations due to cerebral artery occlusion or before hypoxic SD-like depolarizations in rat brain slices [8]. Dreier et al. (2002) demonstrated that ET-1 (10 nM and 1 μM) application induces SD and a rise in [K]o in vivo but not in brain slices, higher concentration (1 μM) of ET-1 led to developing multiple SDs (one to five) [8].

2-DG is a derivative of glucose. It is phosphorylated by hexokinase and forms 2-DG-phosphate, which accumulates in the cell and cannot be used for glycolysis [10]. Glycolysis blockage leads to ATP depletion and alters the extracellular ionic gradient; for instance, it leads to reduced [Ca]o and increased [K]o and glutamate levels [11]. Choki et al. (1984) showed that the rate of K decline (return to baseline) was slower in cases with heterogeneous patterns of low and high glucose metabolism compared to cases with a homogeneous pattern of low glucose metabolism [12]. Both groups showed a similar increase in [K]o during ischemia but different types of response after the release of an ischemic occlusion [12]. There was a positive linear correlation between [K]o half time recovery and glucose utilization, meaning that animals with faster recovery of [K]o demonstrated lower glucose utilization [12]. The slowest recovery of K was seen after ischemia presented the highest rate of glucose utilization [12].

Na/K ATPase pumps out three Na and brings in two K with each translocation cycle, thereby returning [Na]i and [K]o to their resting levels leading to a restored neuronal resting membrane potential after neuronal activity [13]. Na/K ATPase’s activity can be blocked by ouabain—a toxic substance that binds to and inhibits Na/K ATPase [6,14,15]. Ouabain (10 μM) application onto immature rabbit hippocampal slices led to a significant K efflux, longer evoked-SD and higher evoked [K]o response (evoked by injecting 2 M KCl) [15]. Ten to fifteen hours of exposure, 80 μM ouabain induced a massive depletion of intracellular K, which was attenuated by tetraethylammonium in cultured cortical neurons [16]. Five to fifteen minutes after 100 μM ouabain application onto rat hippocampal slices, a sudden negative shift was seen in the DC sustained potential, a shift identical to SD which was accompanied by a markedly increased [K]o. In some slices, after recovery, a second SD-like shift occurred [17].

ATP-sensitive potassium (K-ATP) channels are inward rectifying K channels, regulated by intracellular ATP and ADP levels [18]. High [ATP]i keeps the channel closed while low levels of [ATP]i facilitates activation/opening of the channel [18]. Glibenclamide—a sulfonylurea derivative—is a specific blocker of the K-ATP channel. Glibenclamide application blocked single K-ATP channels in mammalian dentate gyrus granule cells [19] and partially prevented extracellular K accumulation in globally ischemic beating hearts, mainly within the first few minutes of ischemia [20]. Opening of K-ATP channels contributes to extracellular K accumulation during the first few minutes of ischemia [20].

The effect of DNP leading to SD initiation has been reported repeatedly. Although the concomitant increased [K]o has been proposed, the increased [K]o levels and its dynamics have not been hitherto examined. Additionally, the role of ET-1 in SD initiation has been discussed extensively, but the spread of the evoked-K response from a local to a distal site following ET-1 focal application—which is a necessary component of SD—has not been demonstrated. In brief, to address the existing gaps and to study the effect of low [ATP]i on the [K]o, in vivo, we measured neocortical [K]o and the LFP in the presence of DNP, ET-1 or 2-DG, which each reduce [ATP]i through different mechanisms. We found that in vivo [ATP]i depletion in mice neocortex led to increased [K]o along with the concomitant DC-shifts. We proposed that the increased [K]o as a result of decreased [ATP]i is most likely mainly due to the disrupted activity of the Na/K-ATPase pump and the K-ATP channel in the absence of sufficient ATP for their normal activity. To further validate this idea, we blocked the pump and the channel using ouabain and glibenclamide, respectively. We observed an increased [K]o following ouabain usage and a decreased [K]o after glibenclamide usage. Based on the existing literature and experimental evidence presented herein, we propose that [ATP]i and [K]o are inversely related. The present findings allow for a deeper and better understanding of mechanisms involved in [K]o regulation. Furthermore, this study acknowledges one of the consequences of [ATP]i depletion.

## 2. Methods

### 2.1. Animals

Experiments were conducted on 1–2 month-old, 18–30 g CD-1 male mice. Animals were housed in a 12/12 h light cycle with ad libitum access to water and food at Krembil Research Institute. All experimental procedures performed in vivo were approved by the Krembil Research Institute; AUP 3015.20 and 3015.21. DNP was tested in n = 8 for measuring resting [K]o and LFP (which were divided into 2 groups for data analysis) and on n = 7 for measuring resting 2LFP responses. For ET-1 experiments, n = 7 were used for measuring resting [K]o and LFP and n = 5 were used for 2 K recordings. 2-DG was tested in n = 4 for measuring resting [K]o and LFP and on n = 5 for measuring 4AP-evoked [K]o and LFP responses. Effect of ouabain on resting [K]o and LFP was examined in n = 6. Effect of glibenclamide on resting [K]o was measured in n = 11.

### 2.2. Craniotomy

Mice were anesthetized with 5% inhaled isoflurane with oxygen flowing at 1 mL/min to induce anesthesia. During the surgery, inhaled isoflurane was reduced to 1.4–1.8% and oxygen to 0.5 mL/min. Craniotomy was performed with a precision drill, removing a circular region of the skull measuring 5 mm in diameter over the right somatosensory cortex. Phosphate-buffered saline (PBS) was applied over the exposed cortex, filling the cavity of the skull, to prevent tissue damage and dehydration. The animal body temperature was maintained at 37.5 °C using a heating pad. Hind limb withdrawal reflexes and breathing rate were observed at regular intervals throughout the experiment to ensure that the animal remained at a surgical plane of anesthesia. After craniotomy, the mice were transferred to the recording chamber.

### 2.3. Electrophysiology: K-Sensitive Recording Electrode

In order to maintain accurate measurements of K by the K-sensitive electrodes, it is necessary to account for the component arising from an electrical field. To this end, a local reference electrode was used to mitigate distortion of K-sensitive electrode readings. The electrodes were pulled borosilicate capillaries (tip diameter ~1 μm, World Precision Instruments, Sarasota, FL, USA). In the case of the K-sensitive electrode, the interior wall of the capillary was silanized with dimethyldichlorosilane vapor and dried at 120 °C for 2 h [21]. K-sensitive electrodes were filled with K ionophore I-cocktail B (Sigma-Aldrich, Oakville, ON, Canada) at the tip and back-filled with 0.2 M KCl solution [21]. The K-sensitive recording electrodes were then calibrated using different concentrations (0, 2.5, 4.5, 6.5 and 22.5 mM) of KCl solutions. The relationship between the measured voltage and the K concentration of the respective solution was derived using the Nicolsky–Eisenmann equation, which is a commonly used method. The latter calibration lines, which were semi-logarithmic and close to linear, were used to determine [K]o in the brain [21].

### 2.4. Electrophysiology: K-LFP Electrodes

A K-sensitive electrode was coupled with a double-barreled LFP recording electrode, creating a K-LFP recording electrode. The double-barreled electrode was filled with saline and cemented to the K-sensitive electrode such that the distance between the tips of the electrodes was approximately 50 μm apart [22]. First, the K-sensitive electrode was mounted to a head stage of an Axopatch 200B amplifier. A differential reference electrode for the head stage was inserted into a chamber of the double-barreled LFP electrode; the other chamber was used to record the extracellular LFP connected to a head-stage of an Axopatch 200B amplifier. This latter signal was differentially recorded from a common ground wire, attached to the scalp. All amplifiers were then digitized (Digidata 1440, Molecular Devices). LFP and extracellular K signals were low-pass filtered at 5 kHz. Electrodes were lowered into the right somatosensory cortex in steps of 0.1 mm under an Olympus BX-61W1 microscope with 4X PlanN objectives.

### 2.5. Recordings from 2 Sites

In experiments investigating the spreading nature of the response, either 2 K-sensitive recording electrodes or 2 LFP recording electrodes were inserted 2 or 4 mm apart from each other. The microinjection pipette was placed about ~10 μm of one of the recording electrode (local site) and 2 or 4 mm away from the other recording electrode (distal site).

### 2.6. Drugs

DNP (used at 1 mM) purchased from Sigma-Aldrich, ET-1 (used at 1 μM), 2-DG (used at 8 mM), ouabain (used at 15 μM) and glibenclamide (used at 5 μM) were purchased from Tocris Bioscience. Focal application of the drug was performed by 3 repetitions of microinjections each lasting 5 ms (PicoSpritzer III, Parker), resulting in a total injection volume of about 1 μL. Topical application was performed by applying 10 μL of the drug solution on the exposed brain covered by a piece of kimwipe.

### 2.7. Statistical Analysis

Data analysis was performed using MATLAB R2021a (9.10.0.1649659) and Microsoft Excel 2018 version 16.16.2 (180910). Amplitude (amp), duration (dur) and response’s lag were measured for all responses. Amplitude of the raised [K]o was calculated by subtracting the peak of the response from the baseline value. Amplitude of the DC-shift was calculated by subtracting the dip of the LFP response from the baseline. Responses’ lag is the time required to see onset of the responses after drug application. Results were reported as group average +/− SEM. A two-tailed hypothesis was tested and the *p*-value was calculated using t-Test: two-sample assuming unequal variances. To flag the levels of significance a *p*-value less than 0.05, 0.01, 0.001 are summarized with *, **, ***, respectively.

In the control condition, only aCSF was applied and changes in the [K]o and DC-shifts amplitude were calculated by subtracting the [K]o or DC-shifts after aCSF application from the baseline in that animal. Only amplitude of the control and the treatment condition were compared for calculating the level of significance. In the control condition, the peaks or dips were not observed, so there was no duration or response lag associated with the control to be used for comparison purposes. In fact, the changes in the [K]o and LFP were solely due to drug application.

## 3. Results


**DNP application resulted in an overall increase in [K]o along with DC-shifts:**


Topical application of 1 mM DNP on mouse brain resulted in an overall increase in [K]o and concomitant DC-shifts (Figure 1 and Figure 2) recorded using K sensitive and LFP recording electrode (Figure 1A and Figure 2A) The increased [K]o was evident in two different regimes. In the four animals, 0.6 +/− 0.1 min after DNP topical administration (responses’ lag), a single long-lasting increase in [K]o with an amplitude of 12 +/− 2 mM (*p* < 0.01) and duration of 10.1 +/− 1.6 min was evident, which was accompanied by a DC-shift (−10.8 +/− 1.9 mV, *p* < 0.01) lasting for 9.8 +/− 2 min (Figure 1B). In the other four animals, the increased [K]o manifested as two to five recurrent raised [K]o responses (Figure 1C, top). The first, second, third and fourth recurrent responses (coupled K-LFP) were observed 0.7 +/− 0.1 min, 10.9 +/− 1.3 min, 15.0 +/− 2.6 min and 17.0 +/− 3.0 min, respectively, after the initial topical application of DNP (named response’s lag) (Figure 1C, bottom). Since the number of recurrent events between the animals was variable, in each animal only the largest raised [K]o was taken into account for calculating the average amplitude of the four animals. The amplitude of 15.3 +/− 1.5 mM was calculated by subtracting the peak amplitude from the baseline [K]o which was significantly different from the control (Figure 1C, bottom (*p* < 0.01). LFP responses presented the same pattern as [K]o; only the deepest DC-shift in each animal was considered in the data analysis which was −14.8 +/− 2.5 mV (Figure 1C, bottom (*p* < 0.01). Each recurrent raised [K]o event and DC-shift lasted for 1.7 +/− 0.1 min and 1.7 +/− 0.3 min, respectively (average of all recurrent events, N = 14 events, from four animals; data not shown). In three cases, the last event was quite prolonged (several minutes) which was not included in the calculated response durations. In Figure 1C, we presented the values for the first four events, not including the last prolonged responses (all four animals showed the first and second event, three of them showed a third event, and two of them showed a fourth response). The spread of the SD-like event was tested in seven animals (Figure 2). To test the spreading nature of the SD-like event triggered by DNP, 1 mM DNP was applied focally closer to one of the recoding electrodes (local site). The DC-shift was recorded both locally and approximately 2 mm away (distal site) from the injection site (Figure 2A). 0.3 +/− 0.0 min after focal application, a DC-shift was observed in the local site with an amplitude of −15.1 +/− 1.2 mV (*p* < 0.001) lasting for 5.4 +/− 1.0 min. 0.6 +/− 0.1 min after DNP application, a DC-shift with an amplitude of −10.5 +/− 1.4 mV (*p* < 0.001) lasting for 5.3 +/− 0.9 min was recorded distally (Figure 2B).


**ET-1 application resulted in increased [K]o along with DC-shifts:**


After a time of 1.0 +/− 0.2 min post topical application of 1 μM ET-1, [K]o increased by 15.9 +/− 1.8 mM (*p* < 0.001) from the baseline [K]o which lasted for 3.7 +/− 0.6 min. This increase was companied by a DC-shift with an amplitude of −24.4 +/− 2.6 mV (*p* < 0.001) lasting for 3.1 +/− 0.6 min (Figure 3A, n = 7). Next, to confirm the spread of the K response from a local to a distal site—which is a necessary component of SD—we focally injected ET-1 closer to one of the recording electrodes (local site) and recorded the response both locally and 4 mm away from the injection site (distal site) (Figure 3C). In the local site, 1.1 +/− 0.3 min (responses’ lag) after ET-1 focal application, [K]o increased to 17.6 +/− 1.7 mM from the baseline which lasted for 3.0 +/− 0.4 min (Figure 2D, n = 5). In the distal site, 2.3 +/− 0.5 min after ET-1 focal application, the [K]o increased to 14.0 +/− 2.2 mM which lasted for 3.1 +/− 0.5 min (Figure 3D).


**2-DG application resulted in an overall increase in [K]o:**


Topical application of 8 mM 2-DG, after 0.5 +/− 2.0 min, increased the [K]o by 4.0 +/− 0.5 mM (*p* < 0.01) lasting for 7.8 +/− 2.9 min (Figure 4(Aa)). Then, the [K]o decreased not significantly by 0.7 +/− 0.2 mM (*p* > 0.05) in three of the four animals (Figure 4(Ab)), after which [K]o slowly increased (for 36.0 +/− 10.1 min) to 4.8 +/− 0.7 mM (*p* < 0.01) (Figure 4(Ac)). The concomitant LFP response showed a small DC-shift (−5.7 +/− 0.6 mV) early in the recording (Figure 4A, bottom, n = 4). Then, the effect of 2-DG on evoked K responses was tested (Figure 4D). To induce increased [K]o, 200 μL of 5 mM 4-aminopyridine (4AP) solution was applied topically on the brain. To validate the efficacy of the 4AP application on the brain tissue, LFP electrodes were used to detect the corresponding epileptiform activity. At the peak of the epileptiform activity, [K]o increased by 9.7+/− 1.0 mM (*p* < 0.001) from the baseline (Figure 4C, n = 5). At this time 2-DG (250 mg/Kg) was administered IP (Figure 4D). 2-DG application reduced the epileptiform activity, which we have previously discussed [23]. One would expect to see the [K]o return to baseline (2.8 +/− 0.1 mM) after seizure activity blockage following 2-DG application. Although the [K]o decreased from 9.7 +/− 1.0 mM to 6.6 +/− 0.9 mM after 2-DG application, it did not return to the baseline level and its decrease was not significant (*p* > 0.05) while the [K]o level after 2-DG application (6.6 +/− 0.9 mM) was still significantly higher that the baseline (*p* < 0.01) (Figure 4C). This suggests that 2-DG inhibited the seizure activity with a separate mechanism of action for increasing [K]o. Blockage of epileptiform activity should return the seizure-induced raised [K]o to the baseline, whereas 2-DG itself tends to increase the [K]o, leading to a moderately persistent raised [K]o post-seizure.


**Ouabain application led to increased [K]o and concomitant negative DC-shifts:**


Topical application of 15 μM ouabain on mouse brain led to increased [K]o along with concomitant negative DC-shifts (Figure 5, n = 6). The increased [K]o was manifested as 2 to 4 recurrent peaks (Figure 5, top). The first, second, third and fourth recurrent responses (coupled K-LFP) were observed 1.2 +/− 0.3 min, 8.4 +/− 1.6 min, 14.3 +/− 3 min and 21.9 +/− 5.8 min, respectively, after the initial topical application of ouabain (Figure 5, bottom). All six animals showed the first and second event, five of them showed a third event, and three of them showed a fourth response. Since the number of recurrent events between the animals was variable, in each animal only the largest raised [K]o was taken to account for calculating the average amplitude of the six animals. The reported amplitude of 11.4 +/− 2 mM (*p* < 0.01) is the difference between the baseline [K]o and the peak of the response (Figure 5A, bottom). LFP responses presented the same pattern as [K]o; only the deepest DC-shift in each animal was considered in the data analysis which was −17.8 +/− 2.7 mV (*p* < 0.01, Figure 5A, bottom). Each recurrent raised [K]o and DC-shift lasted for 2.2 +/− 0.3 min and 1.9 +/− 0.4 min, respectively, (average of all recurrent events (20 in total) in all six animals; data not shown).


**Glibenclamide application led to a decreased [K]o:**


After 1.0 +/− 0.2 min post topical application of 5 μM Glibenclamide (responses’ lag), the [K]o amplitude decreased by 1.0 +/− 0.1 mM (*p* < 0.001) as measured in 11 animals (Figure 6A,B). The [K]o came back to baseline after 15.2 +/− 2.3 min. In 6 of the 11 animals, the decreased [K]o recovered (one to three times) transiently (attempted to reach the baseline level but failed to remain there) (Figure 6B). The other five animals presented a smooth fall and recovery of [K]o (Figure 6A). The deepest point of the falling [K]o was used to calculate the average for 11 animals.

## 4. Discussion

Inter-neuronal and neural to glia communication rely on releasing K into the extracellular space [24]. [K]o regulation is thus central to the maintenance of healthy brain function. To better overcome disorders involving abnormally high levels of neocortical [K]o, it is critical to unravel mechanisms involved in [K]o regulation. Knowing that astrocytic gap junctional coupling and astrocytic membrane potential play only partial roles in [K]o regulation [21,22], we examined the role of another candidate—[ATP]i depletion—in this complicated yet fascinating topic. [ATP]i depletion was attained via collapsing the proton motive force by using DNP, vasoconstriction by using ET-1, or inhibiting glycolysis by using 2-DG. Our findings suggest that an increased [K]o is a consequence of conditions leading to [ATP]i depletion. The raised [K]o is accompanied by DC-shifts—which resemble SD-like depolarizations—and possibly with other neurological complications that are associated with abnormally high levels of [K]o and low levels of ATP.

Metabolic inhibitors including oxidative metabolism un-couplers (DNP and ET-1), glycolysis inhibitors (2-DG) and Na/K ATPase blockers (ouabain) can initiate SD [25,26] and SD is always companied by abnormally increased [K]o. Na/K ATPase activity maintains membrane potential, so inhibition of this pump or inhibiting its metabolic substrate (ATP) results in membrane depolarization [25]. We propose that the increased [K]o following DNP, ET-1, or 2-DG application is most likely due to Na/K ATPase pump inhibition and K-ATP channel activation caused by ATP depletion. The sharp rise in [K]o accompanied by SD-like events could also be due to depolarization of both astrocytes and neurons. It is also likely that the observed response was due to the direct activation of neuronal channels caused by ET-1, DNP or 2-DG application.

In support of our observations, Russell and Brown (1972) reported decreased intracellular K activity following DNP (1 mM) application onto giant neurons of Aplysia [27]. DNP appears to have the same effect as KCl in initiating SD [5]. The difference is that in the KCl method, the depolarization is attained by [K]o artificially, while DNP pre-treatment leads to K accumulation due to a lowered absorption [5]. According to Godfraind et al. (1970), DNP application onto cortical neurons reduces excitability by raising membrane conductance to K and blocks acetylcholine responses [28]. Application of 0.2 mM DNP on fibers recovering from 5–10 min stimulation significantly reduced the K influx from 22 to 3 pmole/cm^2^ s; the reduction could not be due to a decrease in permeability to K as the K efflux either increased or remained unchanged following DNP application [7]. Kaila and Saarikoski (1981) found out that DNP had a depolarization effect at low concentration (0.05–1 mM). The depolarizing effect of DNP showed similar latency (~5 s) to that caused by increasing [K]o [29]. Interestingly, DNP application on *Ricinus communis* roots within the first hour led to a sharp decrease in K uptake; prolonged exposure led to efflux of K from the roots into the medium [30].

Choki et al. (1984) reported that ischemic brain regions with slow recovery of [K]o are not producing a sufficient amount of energy for Na/K ATPase pump functionality [12]. Possibly energy production was impaired in regions that presented enhanced glucose consumption, resulting in slower recovery of [K]o [12]. Rate of [K]o recovery must be correlated with the amount of ATP (energy) available to the cells since the function of Na/K ATPase and neuronal ionic balance are closely correlated [12]. As oxygen or glucose becomes depleted in ischemic brain tissue, ATP production fails, leading to failure of energy-dependent processes (such as ion pumps) necessary for cell survival. ET-1, which mimics an ischemic state, also seems to disturb Na/K ATPase activity in the CNS [31]. ET-1 (100 nM), similar to arachidonic acid (50 μM) inhibited metabolic trafficking of energy substances (i.e., glucose) in cultured astrocytes [32]. Oliveira-Ferreira et al. (2020) applied ET-1 to the cerebellum which resulted in SDs and showed that the decrease in blood flow was similar to neocortical studies, but the ET-1 threshold for inducing SD was higher in the cerebellum [9]. Part of the complex response of neurons to ischemia is a decline in the ATP available to cells, which in turn suppresses many cellular mechanisms which are dependent on ATP, including the functionality of Na/K ATPase that maintains the normal neuronal membrane potential [17].

Lipton and Robacker (1983) observed that increasing [K]o in the presence of 10 mM glucose was accompanied by an increase in intracellular K concentration in guinea pig hippocampal slices [33]. Na/K ATPase activation by [K]o demands glycolytically generated energy. Therefore, during neuronal activity glucose is needed for maintaining [K]o and clearing the K from the extracellular space, making the brain functionality dependent on glycolysis [33]. Neuronal activation leads to a K-mediated increase in H-2-DG uptake which implies that clearance of high [K]o requires a high energy supply [24]. Frequency of interictal-like epileptiform discharges induced by Cs+ application [11] and 4AP-induced epileptiform [23] activity were reduced after 2-DG application. However, although [K]o reduced, this reduction was not significant and it did not return to the baseline level. This suggests that either seizure reduction via 2-DG application is through mechanisms unrelated to [K]o maintenance or that despite some [K]o reduction due to seizure cessation, it remained raised due to 2-DG-dependent glycolysis blockage leading to lowered Na/K ATPase pump activity as well as K-ATP channel impairment in the absence of glucose. However, this proposal requires further investigation. In brief, 2-DG reduced the 4AP-evoked seizure activity but only partially the 4AP-evoked [K]o. The fact that, in our data, 2-DG had also a later effect (after 30 min) on increasing [K]o is possibly due to its mechanism of action, as glycolysis impairment initially leads to loss of 2ATP and consequently leads to blockade of the cell cycle and cell death.

Xiong and Stringer (2000) also suggested that K redistribution via astrocytes and astrocytic spatial buffering plays only a minor role in [K]o regulation after epileptiform activity in dentate gyrus of hippocampal slices from adult rats [34]. In these experiments, Na/K ATPase appeared to be the main [K]o regulatory mechanism [34]. Not only ouabain, but also Dihydro-ouabain (DHO, 20 μM)—a low affinity analogue of ouabain—when applied onto hippocampal slices, induced small increases in resting [K]o [13]. The half time of [K]o clearance slowed significantly and the amplitude of the train-induced [K]o transient increased; both of which are signs of the reduced activity of Na/K ATPase pump (partial inhibition). DHO led to modest elevation of resting [K]o by about 0.8 mM and transient depolarization followed by hyperpolarization [13]. Increased [K]o leads to a higher glucose utilization/uptake to provide the ATP for Na/K ATPase activity [12,24]. Therefore, we could expect that decreased [ATP]i leads to increased [K]o. It is then logical to suggest that the mechanism by which the [K]o was inversely correlated to [ATP]i was most probably controlled by the Na/K ATPase pump and the K-ATP channel. In studying the correlation between glucose utilization and [K]o in cerebral cortex, we could say that the reason for the relatively slow recovery of raised [K]o (raised K usually took longer than 2 min to return to baseline) is perhaps due to the insufficient energy for a prompt restoration of the cortical ion gradient.

There is an analogy between exposing cells to ouabain (Na/K ATPase blocker) and anoxia (lack of substrate (oxygen) for ATP production), where in both cases, Na/K ATPase will not function properly [17]. If we speculate that the dysfunctionality of Na/K ATPase is the primary cause of SD-like generation, then postponing the negative effect of Na/K ATPase dysfunctionality could delay the anoxic depolarization (which presents the same electrophysiological characteristics and ionic properties as SD). Actually, delaying ATP depletion by incubating brain slices in 25 mM creatine resulted in a significant delay for anoxic depolarization occurrence [17,35]. It was also suggested by Lian and Stringer (2004) that decreased ATP level resulted in inhibition of astrocyte’s metabolism, which increased the susceptibility of neurons to spread depression in adult rats [36]. On the other hand, aCSF containing a nitric oxide-lowering agent combined with ouabain resulted in spreading ischemia [37]. Russell and Brown (1972) also reported decreased intracellular K activity following ouabain (0.2 mM) application on giant neurons of Aplysia [27]. A few minutes after Na/K ATPase inhibition using 100 μM ouabain in rat hippocampal slices, afferent fiber stimulation led to a gradual increased amplitude of evoked population spikes followed by a second population spike in some cases, followed by SD-like events [17].

Wu et al. (1992) activated K-ATP channels using DNP. This activation was characterized by increased time-dependent outward currents at positive potentials [38]. DNP-induced outward currents were completely reversed within 3 min after glibenclamide application in guinea pig ventricular myocytes [38]. In quiescent ischemic heart tissue—where membrane potential is close to equilibrium potential for K—glibenclamide had no effect on K accumulation [20]. In our experiments, K-ATP channel blockage decreased the resting [K]o by about 1 mM. The decrease in [K]o might have been greater if the roles of the K-ATP channels were tested in evoked states rather than resting level. Therefore, further experiments are required to unravel the effect of glibenclamide on evoked K responses.

It might be also important to review the functional interaction between Na/K ATPase pump and K-ATP channel. It is known that Na/K ATPase and K-ATP channel share common modulators [39]. For instance, in rats, skeletal muscle fibers insulin stimulate the Na/K ATPase during hypokalemia, which is counterbalanced by K-ATP channel activation [39]. Insulin stimulating Na/K ATPase activates K-ATP channel [40]. It has been proposed that this activation is facilitated through a membrane-delimited interaction which controls the K homeostasis in skeletal muscles [40]. In fact, the intracellular insulin signal which links the glucose homeostasis to K-ATP channel is the phosphoinositide3-kinase [40].

In the present data, return of the [K]o and DC-shift to the baseline and occurrence of a second, third or fourth event due to DNP and ouabain application, is a representation of cell repolarization or recovery of membrane potential after depolarization; otherwise, one would expect a steady depolarization and steady increase in [K]o. Such a phenomenon at least partly is attributed to Na/K ATPase activity, implying that inducing SD does not require global/complete failure of Na/K ATPase pump. Partial inhibition of Na/K ATPase may initiate slow and progressive intracellular Na and extracellular K accumulation, which by themselves, even before the complete inhibition of the pump, induces SD through not fully understood mechanisms [17]. On the other hand, in the presented data, the pattern of the recurrent events in some of the animals is an indicative of powerful K regulatory/buffering mechanisms attempting to restore the [K]o at the baseline level by clearing the excess amount of K [21]. In our data, the time interval between the first and second recurrent event was much longer than the time between the second and third or the third and fourth events; implying that during the drug exposure, the Na/K ATPase and other [K]o regulatory mechanisms gradually became exhausted, and the tissue became more prone to generate SDs.

In our experiments, the maximum amplitude increase in [K]o due to DNP application was 15.3 +/− 1.5 mM, and due to ET-1 application was 17.6 +/− 1.7 mM, whereas the increase in [K]o caused by ouabain application was 11.4 +/− 2.0 mM, which is smaller than both values seen with DNP and ET-1. This probably means than mechanisms other than Na/K ATPase such as K-ATP channel activity are also involved in the observed raised [K]o. There is a possibility that the concentration of the drugs was not comparable for manifesting the same amount of change in [K]o. It is worth mentioning that the drugs used in this study alter the [ATP]i level only partially and cells have other sources of energy to prevent cell death. We must also keep in mind that these drugs and blocking glycolysis have more global effects than just inhibiting the Na/K ATPase pump and K-ATP channel. So the observed effects could have also been due to other mechanisms triggered by drug application. ET-1 also triggers inter-astrocytic Ca waves which are critical in developing SD [8] as SD is accompanied by Ca flow into neurons [17,35]. Additionally, a rise in membrane conductance is facilitated by increased intercellular-free Ca as a result of slow mitochondrial activity [28]. Ca influx also may lead to mitochondria failure which leads to further ATP depletion and cell death [41].

## 5. Conclusions

In conclusion, we demonstrated that [ATP]i reduction leads to raised [K]o and DC-shifts resembling SD-like events. ATP is needed to establish the concentration gradients of Na and K across the cell membrane, mainly by the Na/K ATPase pump, but also the K-ATP channel. In the absence of a sufficient amount of [ATP]i, Na/K ATPase fails to bring the K ions back into the neuron and the K-ATP channel activates allowing for efflux of K ions, resulting in an increased [K]o. This study highlights the importance of [ATP]i and ATP-dependent pumps and channels in regulating the neocortical [K]o. Much work still needs to be carried out to unravel the exact contribution of ATP in K regulation.

## Figures and Tables

**Figure 1 biomedicines-10-01568-f001:**
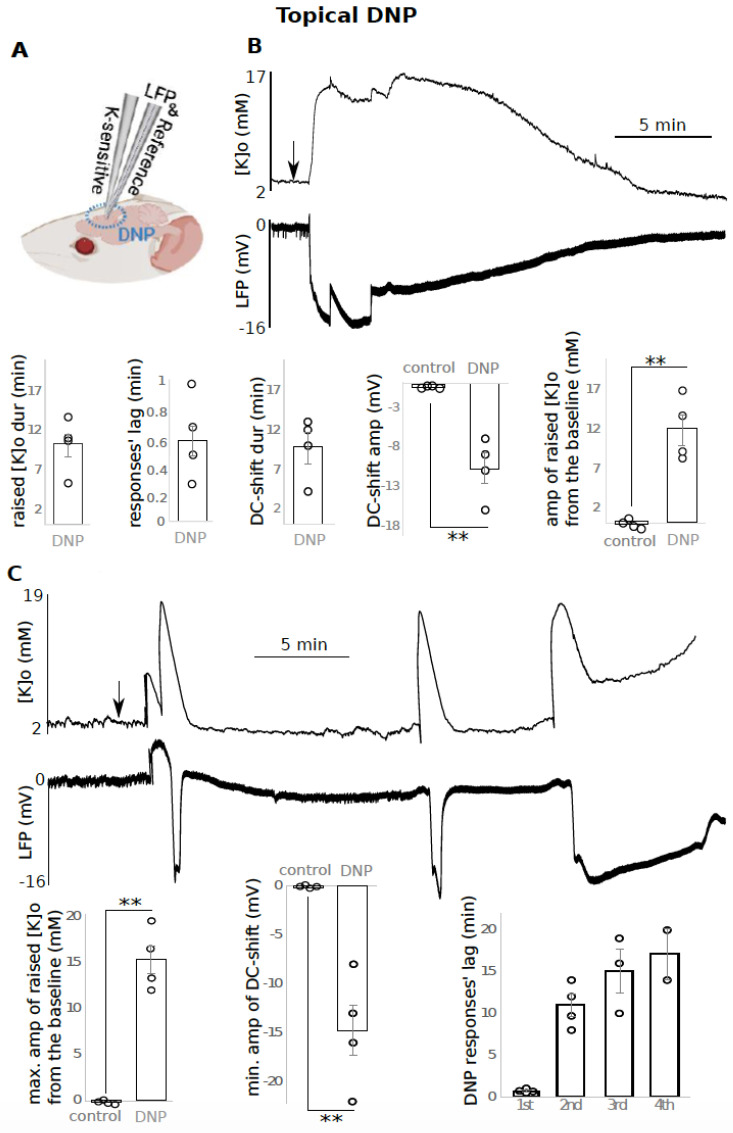
DNP topical application resulted in raised [K]o and negative DC-shifts. (**A**) Experimental configuration for B and C experiments; (**B**) top: a single prolonged raised [K]o and DC-shift (n = 4), bottom: data presentation; (**C**) another type of response after DNP application: top: recurrent peaks in [K]o and dips in the LFP, bottom: data presentation (n = 4), only maximum (max.) of raised [K]o and minimum (min.) of DC-shift amplitude (amp) were included in the average calculation, 2 to 4 recurrent responses occurred per animal. The time between drug application till appearance of each of the recurrent responses is reported as “responses’ lag. Raised [K]o amplitude was calculated by subtracting the peak from the baseline [K]o such that the reported value is the difference. ** indicates significance (*p* < 0.01). dur = duration, amp = amplitude, min = minutes, max. = maximum, min. = minimum. Circle presents individual data.

**Figure 2 biomedicines-10-01568-f002:**
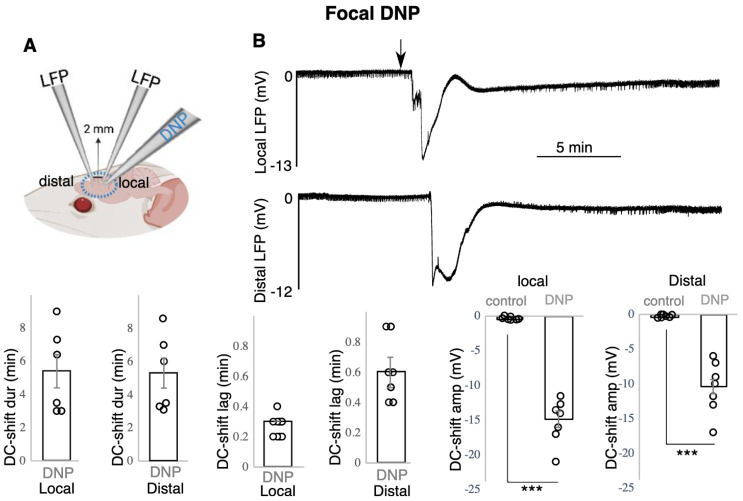
DNP focal application resulted in spreading negative DC-shifts. (**A**) Experimental configuration, (**B**) top: DC-shift spreading from local to distal site following focal application of DNP, bottom: data presentation (n = 7). *** indicates significance (*p* < 0.001), dur = duration, amp = amplitude, min = minutes. Circle presents individual data.

**Figure 3 biomedicines-10-01568-f003:**
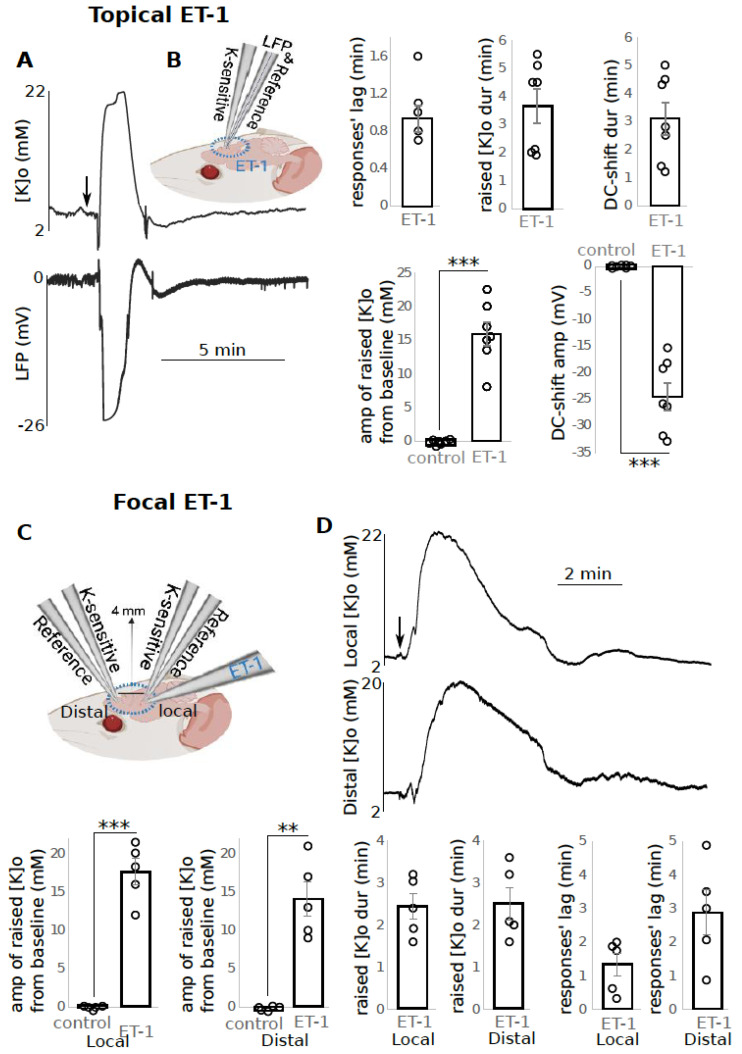
ET-1 application led to raised [K]o and negative DC-shifts. (**A**) Raised [K]o and DC-shift following ET-1 topical application (n = 7) on left and the data shown on the right; (**B**,**C**) experimental configuration; (**D**) raised [K]o response spread from the local to distal site following ET-1 focal application (n = 5), data shown on the bottom. ** (*p* < 0.01), *** (*p* < 0.001) indicates significance, dur = duration, amp = amplitude, min = minutes. Circle presents individual data.

**Figure 4 biomedicines-10-01568-f004:**
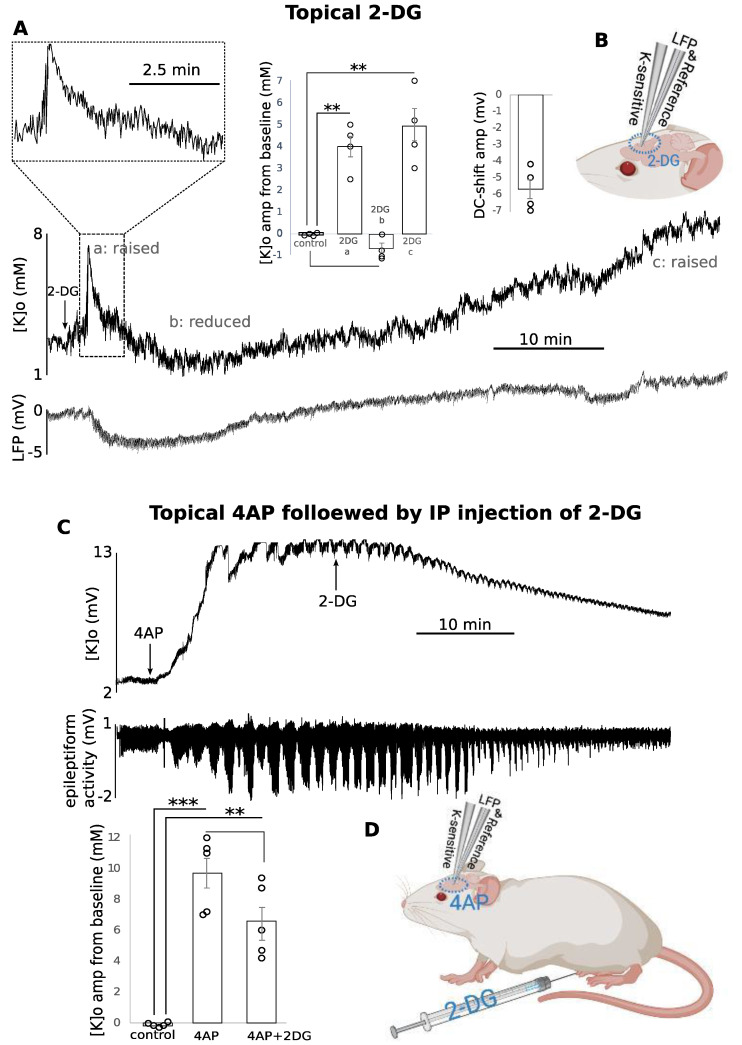
Changes in resting and 4AP-evoked [K]o following 2-DG application. (**A**) Topical application of 2-DG led to an initial increase (a) followed by a slight reduction (b) and then a long and slow increase (c) in the [K]o, and a small DC-shift in LFP was also detected early in the response (n = 4); (**B**) experimental configuration; (**C**) 4AP-evoked raised [K]o (**top**) and epileptiform activity (**bottom**) before and after and 2-DG (n = 5); (**D**) experimental configuration. ** (*p* < 0.01), *** (*p* < 0.001) indicates significance, dur = duration, amp = amplitude, min = minutes. Circle presents individual data.

**Figure 5 biomedicines-10-01568-f005:**
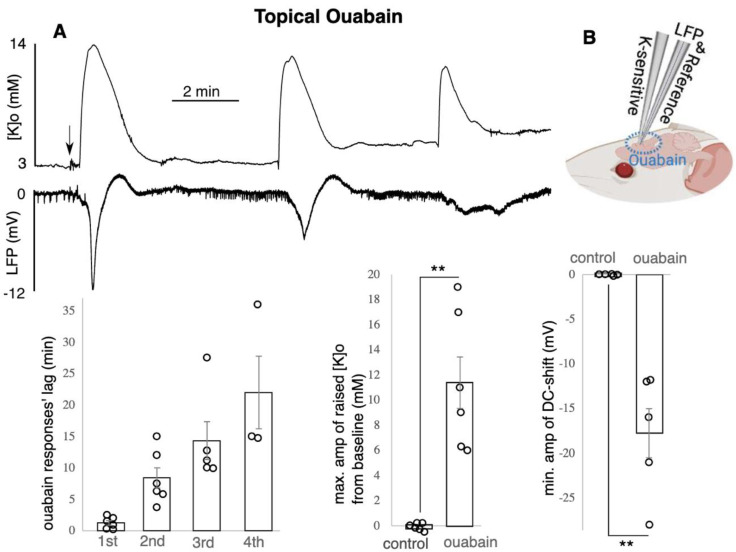
Ouabain led to raised [K]o and DC-shifts. (**A**) Recurrent peaks in [K]o and dips in LFP shown on top and the data shown in the bottom (n = 6), only max. of the raised [K]o and min. of the DC-shift amplitude was included in the average calculation, two to four recurrent responses occurred per animal: the time between drug application till appearance of each of the recurrent responses is reported as “responses’ lag”. Raised [K]o amplitude was calculated by subtracting the peak from the baseline [K]o such that the reported value is the difference. (**B**) experimental configuration. ** indicates significance (*p* < 0.01), amp = amplitude, min = minutes, max. = maximum, min. = minimum. Circle presents individual data.

**Figure 6 biomedicines-10-01568-f006:**
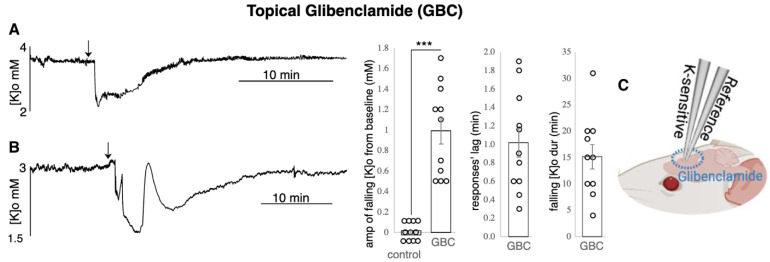
Glibenclamide (GBC) led to a decreased [K]o. The decrease was manifested in 2 types: (**A**) [K]o had a pretty smooth recovery to the baseline (n = 5) and (**B**) [K]o bounced up and down before a full recovery. The min. amplitude of the falling [K]o was calculated by subtracting the dip from the baseline [K]o so the reported value is the difference (n = 6); (**C**) experimental configuration. *** indicates significance (*p* < 0.001), dur = duration, amp = amplitude, min = minutes, max. = maximum, min. = minimum. Circle presents individual data.

## Data Availability

The data presented in this study are available on request from the corresponding author.

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
