# Peer review of "Effects of In Vivo Intracellular ATP Modulation on Neocortical Extracellular Potassium Concentration"

_biomedicines, 2022, doi:10.3390/biomedicines10071568_

Round 1

Reviewer 1 Report

The manuscript submitted by EbrahimAmini et al., titled: "Effect of intracellular ATP modulation on neocortical extracellular [K]o in vivo" is an in vivo study investigating the effects of intracellular ATP modulation to how [K]o is affected.

This is an interesting study with mechanistic insights. The reviewer would like to bring up the following points for the authors to consider:

1. In the methods section please provide more information on the selection of the model. Such as number of mice, power calculation and/or rationale for selecting this particular number, sex, vendor etc. Also please provide the number of the animal protocol approval.

2. A significant portion (the majority) of the literature cited seems fairly old. With the exception of 2 articles that are from the authors (self-reference) that are recent there seems to be a lag in the currency of the literature. This is highly recommended to be addressed especially in terms of the authors discussing their findings in relation to what has already been done and what others have found. In this sense while the reviewer understands the results and how they are presented the discussion needs to be significantly enriched and diversified in the regard discussed above.

Reviewer 2 Report

The manuscript is of interest to the readers but  there are some concerns:

Minor

The manuscript requires editorial rev for instance the special character u in place of the micro along with the text

Figure 4 revises the title

mayor

general comment: do you have any changes in DC shift following topical application per se in the absence of drugs? Please explain the methods if any or not

page 3: the in vivo animal study is not fully described “All experimental procedures performed in vivo were approved by the Krembil Research Institute” is not sufficient. According to the EU rule on the animal investigation, the authors need to report the code and the date of the approved protocol.

Discussion : The role of the KATP channels in regulating extracellular K ion concentrations and of the Na/K ATP ase is indeed well established in Musculo Skeletal apparatus where it plays a role in secondary and primary hypokalemic periodic paralysis in animals and humans for instance and the related references need to be reported (see  The biophysical and pharmacological characteristics of skeletal muscle ATP-sensitive K+ channels are modified in K+-depleted rat, an animal model of hypokalemic periodic paralysis. Tricarico D, Pierno S, Mallamaci R, Brigiani GS, Capriulo R, Santoro G, Camerino DC. Mol Pharmacol. 1998 Jul;54(1):197-206. doi: 10.1124/mol.54.1.197) . The Na/K ATP ase and KATP channels share common modulators for instance insulin stimulates the Na/K ATP ase with hypokalemia that is counterbalanced  by the activation of the KATP channels in muscle
(Modulation of ATP-sensitive K+ channel by insulin in rat skeletal muscle fibers. Tricarico D, et al., Biochem Biophys Res Commun. 1997 Mar 17;232(2):536-9. doi: 10.1006/bbrc.1997.6320.). Furthermore, functional interaction between Na/K ATP-ase and KATP channel has been also reported and needs to be circumstantiated in the discussion section of this manuscript reporting also related references (Involvement of 3Na+/2K+ ATP-ase and Pi-3 kinase in the response of skeletal muscle ATP-sensitive K+ channels to insulin. Tricarico D, Montanari L, Conte Camerino D. Neuromuscul Disord. 2003 Nov;13(9):712-9. doi: 10.1016/s0960-8966(03)00095-6.).

 The reference list needs to be revised as suggested before

Reviewer 3 Report

Authors present an in vivo animal study on mice where, using electrophysiological methods with direct potassium measurment following craniotomy,   the effect of decreased intracellular ATP concentration ([ATP]i) on the extracellular potassium ion concentration ([K]o, in neuronal and glial cells was investigated. Reducing [ATP]i led to raised [K]o and DC-shifts resembling spreading depolarization-like events. The authors propose the following mechanism: increased [K]o is due to the impairment of the Na/K ATPase pump and the ATP-sensitive potassium channel in the absence of sufficient ATP because, Na/K ATPase inhibition led to increased [K]o and ATP-sensitive potassium channel impairment resulted in decreased [K]o. iATP was modulated using DNP, ET-1, 2-DG, Oubain and Glibenclamide. 

Introduction is well written and gives the necessary information on the ratio of the study and current knowledge and concepts. Role of ATP depletion on extracelullar neocortical potassium levels has not been sufficiently investigated; one potential point of criticism is that the authors look somewhat isolated at the effect of ATP depletion in trying to describe the mechanism of maintaining the extracelullar potassium concentration, which is essential for inter-neural and glial communication and healthy brain functioning. In Materials and methods I suggest to include which mice/rats were used, how many animals were sacrified for which experiment. Furthermore, it should be clearly state if neurons, glial or both cell families underwent experiments. 

DNP, ET-1, or 2-DG lead to Na/K ATPase pump inhibition and K-ATP channel activation due to ATP depletion, which leads to increase in extracelullar Kalium. However, authors do not discuss if there are possible secondary mechanisms which each of these substances can cause, aside of ATP depletion, which can lead to extracelullar potassium increase; are there potential neurotoxic effects, desintegration of surrounding glial and neuronal cells which lead to increase of ATP; change of the pH of the microsurrounding or other mechanisms. Also the literature which the authors cite is 30-40 years old; this implies that there have not been similar works in the recent literature - which makes one suspicious, if these mechanisms have already been defined and explained elsewhere. I suggest to include the current knowledge -aside from the point of view of the authors  - on regulation of potassium metabolism in the brain. Also, there are multiple mechanisms, channels and receptors which could play an important role. I suggest to include the following manuscript for citation and I suggest the authors to extensively comment on the role of Kv channels:

Higerd-Rusli GP, Alsaloum M, Tyagi S, Sarveswaran N, Estacion M, Akin EJ, Dib-Hajj FB, Liu S, Sosniak D, Zhao P, Dib-Hajj SD, Waxman SG. Depolarizing NaV and hyperpolarizing KV channels are co-trafficked in sensory neurons. J Neurosci. 2022 May 18:JN-RM-0058-22. doi: 10.1523/JNEUROSCI.0058-22.2022. Epub ahead of print. PMID: 35589395.

In which type of cells were the experiments conducted, since there is obviously a major difference between sensory and motor neurons? Could the role of the neurons play a role in the regulation of K(o), are there any differences between different regions of the brain? 

What are the consequences of your conclusions, future directions, potential therapeutic advantages? What effect do other K-channels and mechanism impose on extracellular K-concentration. It seems that the authors are overly concentrated on only two types of channels - Na/K pump and ATP-K channels while ignoring other possible mechanisms, which can lead to bias. The authors exibit also a vast amount of self-citation, which is sometimes inevitable, but makes one question of the reproducibility of the results. Were there any differences in the effects between the 5 substances the authors have used for experiments? Could these substances cause any secondary effects which lead to ATP depletion and increase in K(o) aside of the mentioned?

What could be the role of the effects on vascular system and M-channels and M-current? Are there any interchangeble relations to intracellular calcium? : 

Ando K, Tong L, Peng D, Vázquez-Liébanas E, Chiyoda H, He L, Liu J, Kawakami K, Mochizuki N, Fukuhara S, Grutzendler J, Betsholtz C. KCNJ8/ABCC9-containing K-ATP channel modulates brain vascular smooth muscle development and neurovascular coupling. Dev Cell. 2022 Jun 6;57(11):1383-1399.e7. doi: 10.1016/j.devcel.2022.04.019. Epub 2022 May 18. PMID: 35588738.

I suggest to thoroughly revise the manuscript and include all the current data on potassium metabolism in the brain. Including a scheme or a diagram of the current knowledge and proposed mechanism of the authors would increase the understanding of the study results:

K-TREK channels and the role of glutamate:

Ávalos Prado P, Chassot AA, Landra-Willm A, Sandoz G. Regulation of two-pore-domain potassium TREK channels and their involvement in pain perception and migraine. Neurosci Lett. 2022 Mar 16;773:136494. doi: 10.1016/j.neulet.2022.136494. Epub 2022 Jan 31. PMID: 35114333.

Li F, Zhou SN, Zeng X, Li Z, Yang R, Wang XX, Meng B, Pei WL, Lu L. Activation of the TREK-1 Potassium Channel Improved Cognitive Deficits in a Mouse Model of Alzheimer's Disease by Modulating Glutamate Metabolism. Mol Neurobiol. 2022 Jun 9. doi: 10.1007/s12035-022-02776-9. Epub ahead of print. PMID: 35678977.

Slick, sodium-activated K-channels:

Flauaus C, Engel P, Zhou F, Petersen J, Ruth P, Lukowski R, Schmidtko A, Lu R. Slick Potassium Channels Control Pain and Itch in Distinct Populations of Sensory and Spinal Neurons in Mice. Anesthesiology. 2022 May 1;136(5):802-822. doi: 10.1097/ALN.0000000000004163. PMID: 35303056.

Effects of elevated extracellular potassium:

Raiteri L, Zappettini S, Milanese M, Fedele E, Raiteri M, Bonanno G. Mechanisms of glutamate release elicited in rat cerebrocortical nerve endings by 'pathologically' elevated extraterminal K+ concentrations. J Neurochem. 2007 Nov;103(3):952-61. doi: 10.1111/j.1471-4159.2007.04784.x. Epub 2007 Jul 27. PMID: 17662048.

Which role does the hypoxia play in the K(o) und could it be possibly more important than the investigated mechanisms? Are there any differences between different parts of the brain, as suggested: 

Farhat E, Devereaux MEM, Cheng H, Weber JM, Pamenter ME. Na+/K+-ATPase activity is regionally regulated by acute hypoxia in naked mole-rat brain. Neurosci Lett. 2021 Nov 1;764:136244. doi: 10.1016/j.neulet.2021.136244. Epub 2021 Sep 14. PMID: 34530116.

In-vivo studies cannot be done in many ways, i.e. craniotomy is needed in order to perform in-vivo electrophysiologic measurment. However, the potential role of this traumatic event on results has not been discussed. Head trauma and recurrent concussion can lead to impairment of the Na/K pump to keep up with the ionic gradient - this has been proposed as one of the mechanisms of cognitive impairment. In the light of these findings, please comment:

Cassol G, Cipolat RP, Papalia WL, Godinho DB, Quines CB, Nogueira CW, Da Veiga M, Da Rocha MIUM, Furian AF, Oliveira MS, Fighera MR, Royes LFF. A role of Na+, K+ -ATPase in spatial memory deficits and inflammatory/oxidative stress after recurrent concussion in adolescent rats. Brain Res Bull. 2022 Mar;180:1-11. doi: 10.1016/j.brainresbull.2021.12.009. Epub 2021 Dec 23. PMID: 34954227.

; which role could the fact that the measurments were performed on a freshly craniotomized mice influence the K-efflux? 

For future directions and applications in humans, I suggest to cite and give your personal opinion and possible use of your results: 

Papadimitriou KI, Wang C, Rogers ML, et al. High-Performance Bioinstrumentation for Real-Time Neuroelectrochemical Traumatic Brain Injury Monitoring. Front Hum Neurosci. 2016;10:212. Published 2016 May 19. doi:10.3389/fnhum.2016.00212

What are your thoughts on methods of measurment of the K(o) and its effect on the Results. One recent study has questioned the reliability of these measurments:

Shen M, Pan T, Ning J, Sun F, Deng M, Liao J, Su F, Tian Y. New nanostructured extracellular potassium ion probe for assay of cellular K+ transport. Spectrochim Acta A Mol Biomol Spectrosc. 2022 May 28;279:121435. doi: 10.1016/j.saa.2022.121435. Epub ahead of print. PMID: 35653810.

You investigate the effects of the two mechanisms - Na/K pump and ATP-K channels. It has been postulated that 50% of the brain energy goes on Na/K pump functioning. In percentage, what is your guess on which of these both mechanisms is more important, i.e. are there potential compensations in the case of traumatic brain injury, ischemia etc. :

Kalocayova B, Snurikova D, Vlkovicova J, Navarova-Stara V, Michalikova D, Ujhazy E, Gasparova Z, Vrbjar N. Effect of handling on ATP utilization of cerebral Na,K-ATPase in rats with trimethyltin-induced neurodegeneration. Mol Cell Biochem. 2021 Dec;476(12):4323-4330. doi: 10.1007/s11010-021-04239-6. Epub 2021 Aug 24. PMID: 34427815.

Furthermore, in order to bring your results in the context of recent literature I suggest next to the above references to cite the following and comment:

Akyuz E, Koklu B, Uner A, Angelopoulou E, Paudel YN. Envisioning the role of inwardly rectifying potassium (Kir) channel in epilepsy. J Neurosci Res. 2022 Feb;100(2):413-443. doi: 10.1002/jnr.24985. Epub 2021 Oct 29. PMID: 34713909.

Muntean BS, Marwari S, Li X, Sloan DC, Young BD, Wohlschlegel JA, Martemyanov KA. Members of the KCTD family are major regulators of cAMP signaling. Proc Natl Acad Sci U S A. 2022 Jan 4;119(1):e2119237119. doi: 10.1073/pnas.2119237119. PMID: 34934014; PMCID: PMC8740737.

Murphy JG, Gutzmann JJ, Lin L, Hu J, Petralia RS, Wang YX, Hoffman DA. R-type voltage-gated Ca2+ channels mediate A-type K+ current regulation of synaptic input in hippocampal dendrites. Cell Rep. 2022 Jan 18;38(3):110264. doi: 10.1016/j.celrep.2021.110264. PMID: 35045307.

Author Response

rebuttal letter is attached. 

Round 2

Reviewer 1 Report

The authors have made a reasonable effort in addressing the reviewer’s comments. Proofreading is suggested.

Author Response

Thanks for your comment. As per my discussion with Ms. Hillary Liu I will definitely proofread the final version after the paper got accepted.

Reviewer 2 Report

The manuscript has been improved

minor changes:

Fig 4 panel c : subtitle, please check

Author Response

Thank you for your comment, Caption of Fig.4 is now edited.